# Interactions between Medications and the Gut Microbiome in Inflammatory Bowel Disease

**DOI:** 10.3390/microorganisms10101963

**Published:** 2022-10-04

**Authors:** Julia Eckenberger, James C. Butler, Charles N. Bernstein, Fergus Shanahan, Marcus J. Claesson

**Affiliations:** 1APC Microbiome Ireland, University College Cork, T12 YT20 Cork, Ireland; 2School of Microbiology, University College Cork, T12 TP07 Cork, Ireland; 3The SFI Centre for Research Training in Genomics Data Science, H91 TK33 Galway, Ireland; 4Inflammatory Bowel Disease Clinical and Research Centre, University of Manitoba, Winnipeg, MB R3A 1R9 Canada; 5Section of Gastroenterology, Department of Internal Medicine, University of Manitoba, Winnipeg, MB R3A 1R9, Canada; 6Department of Medicine, University College Cork, T12 AK54 Cork, Ireland

**Keywords:** inflammatory bowel disease, gut microbiota, drugs

## Abstract

In view of the increasing evidence that commonly prescribed, non-antibiotic drugs interact with the gut microbiome, we re-examined the microbiota variance in inflammatory bowel disease (IBD) to determine the degree to which medication and supplement intake might account for compositional differences between disease subtypes and geographic location. We assessed the confounding effects of various treatments on the faecal microbiota composition (16S rRNA gene sequencing) in persons with Crohn’s disease (CD; n = 188) or ulcerative colitis (UC; n = 161) from either Cork (Ireland) or Manitoba (Canada) sampled at three time points. The medication profiles between persons with UC and CD and from different countries varied in number and type of drugs taken. Among Canadian participants with CD, surgical resection and overall medication and supplement usage is significantly more common than for their Irish counterparts. Treatments explained more microbiota variance (3.5%) than all other factors combined (2.4%) and 40 of the 78 tested medications and supplements showed significant associations with at least one taxon in the gut microbiota. However, while treatments accounted for a relatively small proportion of the geographic contribution to microbiome variance between Irish and Canadian participants, additive effects from multiple medications contributed significantly to microbiome differences between UC and CD.

## 1. Introduction

Inflammatory bowel disease (IBD) which comprises Crohn’s disease (CD) and ulcerative colitis (UC) have been linked with changes in the gut microbiome [1,2], although cause and consequence have not been disentangled [3,4]. Moreover, no uniform microbiome pattern or signature has been identified consistently across multiple studies [5]. This may be due to the heterogeneity of IBD and/or variations in study design and patient populations [3]. Medications have also been shown to alter the gut microbiome and add an additional layer of confounding factors. Not only antibiotics have a well-described, cumulative long-lasting effect on the intestinal microbiome [6], but also commonly prescribed non-antibiotic medications have been shown to impact the gut microbiome [7,8]. Standard therapies for IBD include anti-inflammatory drugs, such as 5-aminosalicylic acids (5-ASA) and corticosteroids, and immunosuppressants, such as thiopurines, methotrexate and biologics (TNF-α inhibitors and integrin or interleukin receptor antagonists). Several studies have shown that these drugs can impact the microbial composition [9,10,11,12,13], but vice versa, gut microbiota can modulate their pharmacological activity via drug (in)activation and biotransformation [14]. For instance, bacterial azoreductases release 5-ASA from its prodrugs, while bacterial N-acetyltransferases are responsible for the inactivation of 5-ASA [9]. Aminosalicylates in turn can increase the levels of some Firmicutes while reducing the levels of Bacteroidetes and Proteobacteria [15].

Moreover, commonly used non-IBD drugs can also affect the composition and metabolic function of gut microbiota [16]. In a major in vitro study, Maier and colleagues [8] tested the effect of more than 1000 drugs against 40 representative gut bacterial strains and found that approximately a quarter of human targeted drugs inhibited the growth of at least one test strain, with butyrate and propionate producing bacteria being more sensitive and γ-Proteobacteria being more drug-resistant. Pathogens and commensals alike have been shown to be able to metabolize and/or bio-transform a variety of drugs leading to an altered bacterial metabolism and changes in microbial composition due to metabolic cross-feeding and changes in the intestinal microenvironment [17,18]. To this end, a variety of studies have elucidated the microbiota-altering effects of commonly prescribed medications (proton pump inhibitors [19,20], lipid lowering statins [21], laxatives [22], metformin [23], beta blockers [24], ACE inhibitors [25], and SSRI antidepressants [26]). In some cases, the drugs explained more of the microbiota variability than the disease itself [27,28], which bears the question as to what degree this variability is secondary to the disease, or to the drugs treating it.

We recently showed in a large intercontinental twin city study of the microbiome in IBD that geographic location (Ireland vs. Canada) had a major influence on microbiota variance almost equivalent to that of a diagnosis of Crohn’s disease itself [29]. While the influence of geography may, in part, be due to cultural and ethnic influences, differences in treatment on either side of the Atlantic may also have a contribution. Therefore, the purpose of the present study was to re-examine and disentangle microbiota variance in IBD to determine the degree to which differences in treatment at different locations might account for the apparent geographic influence. The results confirm that the overall trends of microbiota composition and diversity, as previously reported by us, remain different across IBD-subtypes and geographic location. Only a small part of the effect of geographic location is explained by the differences in medication and supplement intake. However, a large proportion of the disease-associated shift in microbial composition between persons with UC and CD can be explained by additive interaction effects from multiple medications.

## 2. Materials and Methods

The V3-V4 16S rRNA amplicon sequences from our previous study, which had been processed in a single laboratory in Cork utilizing the same protocols, were downloaded and pre-processed as previously described [29], with the exception of the taxonomic classification, which was here performed against the SILVA database (v132) [30] within the mothur suite (v1.39.3) [31] utilizing the classify.seqs function with a bootstrap cut-off of 80%. OTUs that fell below that cut-off were assigned as unclassified at that particular rank. Species-level resolution was provided by SPINGO [32] using a similarity score of 0.5 and bootstrap cut-off of 0.8 against the SILVA database (v132). Conflicts between the two methods were resolved by means of BLASTn (v 2.8.1; 10 May 2021) [33]. The raw medication data was classified based on the anatomical therapeutic chemical (ATC) classification system, which hierarchically classifies medications and dietary supplements based on the therapeutic use of their main active ingredient (1st level: anatomical main group; 2nd level: therapeutic subgroup; 3rd level: pharmacological subgroup; 4th level: chemical subgroup; 5th level: chemical substance) [34] and recorded as a qualitative variable. While surgical resection is neither medication nor supplement, it is a common treatment for IBD as 80% of patients with CD require surgery during their lifetime [35]. It also has been shown to significantly alter the microbiome of patients [36] and was therefore included in the analysis as treatment. Long-term dietary habits were captured through frequencies of medium food servings of 157 items via Food Frequency Questionnaires (FFQ) and were summarized into one factor, the Healthy Food Diversity (HFD) Index, as previously described [29,37]. An active state of IBD was defined as a faecal calprotectin measurement of ≥250 µg/g [38].

All statistical analyses were performed in the R environment version 4.1.0 and visualizations were produced with the ggplot2_3.3.5 package [39]. In order to alleviate the constant sum constraint of compositional data [40], zeros were removed from the raw counts via the count zero multiplicative method within the zCompositions_1.3.4 package [41] and subsequently subjected to a centered log ratio (CLR) transform, i.e.; the data were expressed as logarithms of ratios with the geometric mean as denominator using the propr_4.2.6 package [42]. Beta diversity (i.e.; between sample diversity) of the microbiome was evaluated via principal component analysis on Aitchison distances [43] within phyloseq_1.36.0 [44], while vegan_2.5-7 [45] was used for permutational multivariate analysis of variance between groups. Alpha diversity (i.e.; within sample diversity) was calculated with iNEXT_2.0.20 [46]. Differential taxa abundance and effect sizes without adjusting for confounding factors was computed via ALDEx2_1.24.0 [47] using 1000 Monte Carlo samples. Differences in the usage of medications and supplements between the different IBD-subtypes and participants from the two different geographic locations were tested for significance with Fischer’s Exact tests. Distance-based redundancy analysis (dbRDA) was performed with the capscale function in vegan_2.5.7 [45] to evaluate the effect of environmental factors on the medication profiles (Jaccard distances) and gut microbial composition (Aitchison distances). Here, medication profiles and microbial abundances constitute the set of response variables, while the environmental factors represent the predictive variables. The proportion of explained, compared to the total fitted variance indicates how much of the variation between samples is due to differences in environmental factors. Selection of the most relevant species and features in the dbRDA was implemented with the ordiselect function in goeveg_0.5.1 [48]. Differences in numbers of used medications as well as the number of changed medications and intra-personal differences in alpha and beta diversity measures were assessed via Wilcoxon rank sum tests. 

Explained variance of single covariates in a multivariate data set was computed with the VpThemAll_0.0.0.9 wrapper [49], which chooses a model via the ordistep function that explains most variance in microbiota composition and then looks at each included metadata variable separately, conditioning out the effect of all other included metadata variables using the varpart function. The difference between the naïve (shared) effect and the unique effects amounts to the interaction effect. The variation in community data with respect to explanatory tables was calculated with the varpart function in vegan_2.5-7 [45] allowing only permutations within the samples of the same patient to adjust for multiple measurements. Significant associations and effect sizes of covariates with single taxa were computed with the metadeconfoundR_0.2.8 package [50], specifying “patient” as random variable; *p*-values were adjusted for multiple testing where appropriate, using the Benjamini and Hochberg method [51].

## 3. Results

After extracting from the Clooney et al. [29] data set samples of persons with IBD with all three time points and no missing data, 349 persons were included for further analysis, of which 188 were diagnosed with CD and 161 with UC. The sampling time points were on average 15.18 ± 0.42 weeks apart. Usage information on medication and dietary supplements had been collected for all of the 1047 samples. Of the original 3148 operational taxonomic units (OTUs) that were clustered at ≥97% identity, 2409 remained after filtering, leaving on average 22,024 ± 492 usable reads per sample. Applying the ATC classification system [34] to the raw medication data yielded 302 different chemical substances (5th ATC level). A number of dietary supplements (n = 74) had no ATC classifier and were aggregated under “Other supplements”. For the analysis, the medications and supplements were then further combined into 120 pharmacological subgroups (3rd ATC level), and drug usage was recorded as a qualitative yes–no variable. Long-term dietary habits were summarized into one factor, the Healthy Food Diversity (HFD) Index, as previously described [29,37]. An active state of IBD was defined as a faecal calprotectin measurement of ≥250 µg/g [38]. (Figure 1, Appendix A).

### 3.1. Microbiota Composition Is Different Both between IBD Subtypes and Geographic Locations

In contrast to our original study, here we used a ‘compositionally aware’ analysis approach, which more accurately quantifies taxa without the confounding relative effect of total read count per sample [40]. As previously reported [29], alpha (i.e.; within sample) diversities were lower for all persons with CD compared to UC and also for Canadian participants in general (Wilcox *p* < 0.05; Figure 2a,b, Appendix A). Abundances of OTUs were CLR-transformed, and beta (i.e.; between sample) diversity analysis was calculated from Aitchison distances of all OTUs present in at least 10% of the samples. The first two principal component axes captured a much higher proportion of the microbiota variation in the data set (71.6%), whilst showing the same significance (PERMANOVA *p* < 0.05) for disease and location-associated shifts (Figure 2c,d). Differential taxa abundance analysis was carried out using ALDEx2, initially without accounting for any confounders. Here, the OTUs were aggregated based on their highest known taxonomic classification and filtered for an abundance in at least 10% of samples, resulting in 233 tested taxa. Of these, 108 OTUs were significantly different (Wilcox/Welch *p* < 0.05) between persons with UC and CD (Figure 2e) but displayed only weak to moderate effect sizes, ranging from −0.38 to 0.41. *Faecalibacterium prausnitzii*, Ruminococcaceae UCG-002 and *Eubacterium* were the most increased taxa in patients with UC, while *Lachnoclostridium*, *Erysipelatoclostridium* and *Escherichia/Shigella* had the highest abundance in the microbiome of patients with CD (Appendix A). The effect sizes of the 100 taxa that were significantly different (Wilcox/Welch *p* < 0.05) between subjects from Manitoba and Cork were slightly higher than between the IBD-subtypes (Figure 2f), ranging from −0.33 to 0.59. Members of Saccharimonadaceae, *Streptococcus* and *Ruthenibacterium lactatiformans* were the most enriched taxa in the Manitoba cohort, while Lachnospiraceae_UCG-004, *Odoribacter splanchnicus* and Bacteroidales were the most increased taxa in the Irish cohort (Appendix A). Thus, with updated analysis, compared to our previous work [29], the overall trends of microbiota composition and diversity remain different across IBD-subtypes and geographic location.

### 3.2. Higher Usage of Resection, Medications and Supplements among Canadian Participants

To disentangle the effect of medication on microbiota, we first quantified any significant differences in IBD treatments, both dependently and independently of geography. Of the 120 different pharmacological subgroups found in this study, 78 were recorded at least 5 times as “yes” in total and were included in the downstream analysis (default MetadeconfoundR requirement). We also included surgical resection as a treatment. Of these, 28 medications and supplements showed significantly (Fischer *p* < 0.05) different usage between the Irish and the Canadian cohort (Figure 3a). Between persons with either CD or UC, 18 pharmacological subgroups were differentially (Fischer *p* < 0.05) used. Surgical resection was significantly more present in the Manitoba cohort and in persons with CD. Intestinal anti-inflammatory agents, a subgroup which contains ‘locally acting corticosteroids’ and ‘aminosalicylic acid and similar agents’ were the most commonly used treatments, which were over-represented in persons with UC and Irish participants. The most taken supplements were vitamin A and D, which were over-represented in Canadians with CD (Figure 3a, Appendix A). 

Jaccard distance-based redundancy analysis (dbRDA), which is more suitable for quantifying distances between binary data (e.g.; medication A: TRUE vs. FALSE), was used to evaluate the effect of IBD-subtype and geographic location on medication usage patterns. Both the IBD-subtype (F = 33.71; *p* < 0.001) and geographic location (F = 25.96; *p* < 0.001) showed a significant effect, with a CD/UC diagnosis accounting for 26.6% and location for 19.5% of the explained variation in treatments (Figure 3b,c). Surgical resection and vitamin A and D supplementation were most prevalent in Canadian participants, while intestinal anti-inflammatories best distinguished Irish participants with UC from the other cohorts. 

The participant groups from varying locations and with a different disease type not only diverged in their drug usage pattern, but also in the amount of ingested pharmaceutical compounds. Overall, persons with CD took a significantly higher number of supplements and medications (Wilcox *p* < 0.05) than persons with UC. Similarly, Canadian participants used a significantly greater number of supplements and medications than their Irish counterparts (Figure 4a,b). Another source of variation between the groups was the frequency with which specific medications were changed over the course of the study. Only 73 of 349 participants did not change their medication regime during the 6 months of study, while another 93 persons changed five or more medications over the three time points (Figure 4c). Unsurprisingly, due to the higher usage of pharmaceutical products in the Canadian cohort, there were also significantly (Wilcox *p* < 0.05) more changes of medications in this group, while there were no significant disparities between persons with UC and CD within their cohort (Figure 4d).

### 3.3. Polypharmacy Associated with the Disease-Associated Microbiota Shift in Crohn’s Disease

Polypharmacy (PP) is defined as the concurrent and regular use of at least 5 medications and excessive polypharmacy (EPP) as use of 10 or more different drugs. In our study, 120 persons at 233 time points fit in the former category, while 22 persons (42) fit the latter group (Figure 5b). Polypharmacy expectedly increased with age as persons in the EPP and PP group were significantly older than those that took less than five medications on a regular basis (Figure 5c). The microbial composition of persons with EPP, PP and no polypharmacy (NP) showed a gradual shift along the first PC axis (Figure 5a). This corresponded with the shift seen when grouping by IBD-subtype (Figure 2c), which is not surprising as persons with CD used significantly more medications than persons with UC (Figure 4b). Taking multiple medications at the same time can lead to increased side effects and drug interactions that amplify adverse complications [52].

Since the intake of these pharmaceuticals might influence disease-associated microbiome disturbances, we investigated the influence of changes in the treatment regime on the microbiota. We compared intra-individual alpha and beta diversity differences between the first and second, and second and third time points of participants who did change their medications and those who did not (Appendix A). Persons with CD who started new medications between time points displayed significantly higher Shannon diversity differences (Figure 5e) and intra-individual Jaccard distances (Figure 5g). This finding was not reproduced in persons with UC, which might be due to the higher microbial richness compared to persons with CD. Neither disease group showed significant differences in intra-individual species richness (Figure 5d) nor in Aitchison distances (Figure 5f) after starting a new medication. Thus, while presence or absence of certain taxa can be affected by a change in medication, this does not necessarily translate to overall differences in species richness. Additionally, it may have also been important to know when exactly a change in the medication regime has occurred. Unfortunately, this information was not available to us. 

### 3.4. Treatment Explains More Variation in Microbial Composition Than IBD-Subtype

Having established differences in both microbiota and treatment between locations, as well as microbiota–polypharmacy associations, we further assessed the overall effect of these treatments on the gut microbiota in more detail. Thus, we performed multivariate regression analysis of the explained microbiota variance (Aitchison distances) of different variables such as medications, patient demographics, environmental factors, and disease status and activity. The full explanatory regression models contained 16 of those variables, which together explained 7.88% of the total variation in microbial composition (R^2^: 0.788; Figure 6a, Appendix A). This is slightly less than in our original study (9.7%) [29] and is likely due to the fact that here we considered a slightly different number of participants and no control subjects. A conservative estimate of the unique effect of each variable present in the full model was obtained by evaluating all the variables separately, while at the same time partitioning out the effects of all other included covariates. Thus, the difference in explained variation between the naive effect and the unique effect are the interaction effects between the variables. When taken together, all medications explained more microbiota variation than IBD-subtype alone (1.47% vs. 0.25%) but less than surgical resection (1.68%) or geographic location (1.23%). It was also notable that the interaction effects associated with the proportion of variance explained by IBD-subtype were much higher (1.57%) than the interaction effects explained by geographic location (0.50%) (Figure 6a, Appendix A). Treatment (surgical resection and medications combined) explained more variation (3.48%) than all other clinical and environmental factors together (2.40%) (Figure 6b). The effects of disease remission and intestinal anti-inflammatory agents were positively correlated with UC, while surgical resection, drugs for peptic ulcers and gastro-oesophageal reflux disease (GORD) and antacids as well as EPP and insulins and analogues were positively associated with the effect of CD on the gut microbiome (Figure 6c). In concordance to the previous differential taxa abundance analysis (Figure 2e), *Escherichia/Shigella*, *Klebsiella* and *Blautia* were most associated with CD, whilst a high abundance of several *F. prausnitzii* species characterized persons with UC. Irish participants exhibited a higher abundance of Lachnospiraceae UCG-004 than Canadians, who in turn showed a higher proportion of *Escherichia/Shigella* and *Ruthenibacterium lactatiformans* (Figure 6d).

### 3.5. Disease-Associated Bacteria Are Less Affected by Medications Than Health-Associated Bacteria

To further quantify individual drug-microbiota effects, we implemented a univariate statistical approach that identifies naïve associations between taxa and metadata for confounding effects, i.e.; distortions of the association between taxonomic feature and metadata variable which are caused by additional variables. We included 79 treatments, which were recorded at least 5 times as “yes” within the study, as well as 8 additional metadata variables (geographic location, IBD-subtype, disease activity, smoking status, alcohol consumption, HFD-index, years since diagnosis and biological sex). Of these, 49 variables had at least one significant association (FDR < 0.05, Cliff’s Delta > 0.1) with a taxonomic feature in the gut microbiome (Appendix A). 

When clustering the taxa and metadata associations by their effect sizes (Figure 6e), we noticed that the majority of treatments seem to amplify the effect of IBD on the microbiota in so far that many of the associations between treatments and microbiota had the same direction (i.e.; positive or negative) as the associations between IBD and microbiota. As detailed further below, many of the taxa that we found to be increased or decreased in persons with IBD were also correlated with commonly used treatments of these participants. The health-associated taxa *F. prausnitzii*, Lachnospiraceae NK4AI36-group and *Subdoligranulum* [4,5,53,54] were not only depleted in CD compared to UC, but also had the highest number of (negative) correlations with other covariates (21, 20, 17 and 17 associations, respectively). Equally, many taxa that are positively correlated with IBD, such as *Escherichia/Shigella* (n = 18), *Streptococcus* (n = 22), *Klebsiella* (n = 19) and Veillonellaceae (n = 19), as well as bacteria from the oral cavity such as *Rothia dentocariosa* (n = 22), *Fusobacterium nucleatum* (n = 21) or *Oribacterium sinus* (n = 19), were among those that showed the highest number of positive correlations with medication and supplements (Figure 6e). At higher taxonomic ranks, Verrucomicrobiae, Bacteroidia and γ-Proteobacteria were significantly (Fischer *p* < 0.05) less associated with medications than Actinobacteria, Bacilli and Clostridia, among others (Figure 7a, Appendix A). It was also notable that the Clostridia taxa Ruminococcaceae, Clostridiales vadinBB60 group and unclassified Clostridiales had the highest proportion of negative associations. While Lachnospiraceae were relatively unaffected by medications (Figure 7c), they also showed a relatively high proportion of negative associations (Figure 7d, Appendix A).

### 3.6. Additive Effects of Multiple Medications Amplify the Effect of IBD-Subtype on Gut Microbiota

In terms of specific treatment-association to taxa, surgical resection (198 associations), antipropulsives (167) and intestinal anti-inflammatory agents (140) showed the highest number of significant associations. A combination of estrogens and progesterones, chemotherapeutics for topical use, insulins and peripherally acting antiandrenergic agents showed the strongest associations with single taxa (Figure 8a). Of the overall 2754 significant associations between covariates and the taxonomic features, 35% were confounded, i.e.; the effect size value for a metadata/ taxonomic feature pair could be reduced to at least one other covariate while retaining its own significance (Figure 8b). Markedly, the proportion of confounded associations was lower for geographic location (16.8%) but much higher for IBD-subtype (60.4%). IBD-subtype showed interaction effects with either years since diagnosis, geographic location, biological sex or surgical resection, as well as five other treatments (A07E, B03A, A07D, N02A and A12B in Appendix A). Moreover, 2547 of all associations (92.5%) involved taxa that were also significant for the IBD-subtype, whereof 80% were in the same ± direction, thus potentially amplifying the separation of the gut microbiome composition between persons with CD and persons with UC. For example, the immunosuppressant subgroup (L04A), which contains selective immunosuppressants such as mycophenolic acid but also tumor necrosis factor alpha (TNFα) inhibitors and other immunosuppressants such as methotrexate and azathioprine showed significant associations with 124 taxonomic features. Of these, 120 were not only also significant for IBD-subtype but also co-directional. A notable exemption to this exacerbating effect were intestinal anti-inflammatories, a pharmacological subgroup that contains locally acting corticosteroids and aminosalicylic acid and similar agents and shared 129 of its 140 significant associations with IBD-subtype, but none of those were co-directional, suggesting that they counteracted the effect of IBD-subtype towards a healthier microbiome. Other variables that reduced the effect of IBD-subtype were a combination of progesterone and estrogen (54 shared, whereof only 4 were co-directional), HFD-index (106 shared, 6 co-directional) and biological sex (26 shared, 3 co-directional; Figure 8c). The amplifying effect for geographic location was much less pronounced, as 2037 associations (80%) were shared with other variables and only 60% of those were co-directional. Surgical resection, Vitamin A/D supplementation and antipropulsives had the most co-directional associations with geographic location (111, 92 and 85, respectively). Years since diagnosis and IBD-subtype also showed a high overlap with the effect of geographic location (107 and 109 co-directional; Figure 8d). While most of these effects were weak or moderate (Figure 8a), the summation of the many co-directional effects from different treatments might obscure which taxa were most depleted or increased due to a variable of interest in this case IBD-subtype and to a lesser degree geographic location.

## 4. Discussion

In our previous study, we showed that geographic location accounted for the second highest explained variance in gut microbial composition after a diagnosis with CD [6]. The present study extends the earlier observations by addressing differences in medication profiles in greater detail. We discovered that multiple medications and supplements were differentially used between persons with UC and CD as well as between Canadian and Irish participants. Indeed, Canadians were found to take significantly more medications and supplements than their Irish counterparts, and IBD-subtype accounted for only slightly more variation in drug usage patterns than geographic location. Despite this, when assessing the confounding effects of treatments on the microbiota, only a small part of the variation in microbial composition between participants from the different geographic locations was explained by the differences in medication and supplement intake.

In contrast, a major part of the disparity between the gut microbiomes of persons with UC versus CD seems to be due to, or amplified by, interaction effects with treatment. About half of the tested medications and supplements showed significant associations with at least one taxon from the gut microbiota, and together, treatments, including surgical resection, and medications and supplements, explained more variation in gut microbial composition than all other tested environmental factors. 

Several taxa whose increase are generally reported with a shift away from healthy gut microbiome composition to an inflamed state, including *Escherichia/Shigella*, *Streptococcus*, *Klebsiella* and *Veillonellaceae* [3,4,55,56] were found here to be increased in the microbiome of persons with CD. These taxa also notably ranked amongst the highest number of positive associations with the tested covariates. Bacteria from the oral cavity such as *Rothia dentocariosa* and *Fusobacterium nucleatum*, which have been reported to be increased in PPI users [19,20] showed not only significant positive associations with drugs for peptic ulcers and GORD in the present study but also belong to the top 40 most affected bacteria. A depletion of *Faecalibacterium prausnitzii*, was in this and other studies associated with CD [5,29,57,58] and was also described in persons treated with the immunosuppressant azathioprine [59]. Whilst we were unable to confirm a negative association of *F. prausnitzii* with the intake of immunosuppressants, it was among the taxa which had the most negative associations with the tested drugs. The lack of diminution of this taxon by immunosuppressants might be explainable due to the fact that the ATC subgroup L04A not only includes thiopurines, but also TNFα inhibitors, the latter of which have been shown to increase SCFA producing bacteria [60]. It is notable though that immunosuppressants share nearly all their significant associations to taxonomic features with IBD-subtype, and all of those shared associations showed the same directionality and thus increase the disparity between the gut microbiota of persons with UC and CD. In contrast to that, intestinal anti-inflammatory agents are among the few drugs that did not follow this exacerbating trend. While this medication subgroup also shared most of its significant associations to taxa with IBD-subtype, none of them were co-directional. This observation is in agreement with earlier reports that 5-ASA drugs can partially recover the gut microbiome to a healthy status [29,61]. 

Comparing the distribution of the number of reads and significant associations to medications for each taxon showed that some taxa were more resistant towards the effect of human targeted drugs, e.g.; Verrucomicrobiae, Bacteroidia and γ-Proteobacteria, while others, such as Actinobacteria, Bacilli and Clostridia, were more sensitive. These results concurred with a study from Maier et al. which found γ-Proteobacteria to be more drug resistant than highly abundant commensals such as *Roseburia intestinalis*, *Eubacterium rectale* and *Blautia obeum* [8].

While this study could not confirm treatment as a major factor explaining dissimilarities in gut microbiota of persons from different geographic locations, it shows that the highly variable medication profiles of persons with IBD and their effect on the faecal gut microbiota likely impede the discovery of a universally valid microbial signature distinguishing the IBD-subtypes and at least in part, explain the high disparity between different IBD microbiome studies. Furthermore, it highlights the need to include an exhaustive list of medication intakes (and ideally dosages) of study participants in the analysis that go beyond the common IBD therapeutics to improve reproducibility between IBD-studies.

There are several limitations to our study. Due to the highly variable drug usage patterns, not all medications could be assessed for univariate analysis. The evaluation of the confounding effects of medication on gut microbiota was further hampered by extreme polypharmacy and a multitude of changes in medication regimes over the course of the study. It is therefore possible that some of the reported effects of particular medications were under-estimated. However, while the exact effect sizes of single medications may not be robust, taken together they reveal a trend that treatments can exacerbate disease-associated shifts by additive effects of multiple medications on the microbiome as well as by affecting some groups of microbiota more than others. It is noteworthy though, that all described effects between medication and microbiome in this study are associative rather than causative, and thus intervention studies with treatment naïve persons with IBD and animal models will be needed to comprehensively disentangle the role of treatment and disease on the variation the gut microbiome of patients with IBD.

## Figures and Tables

**Figure 1 microorganisms-10-01963-f001:**
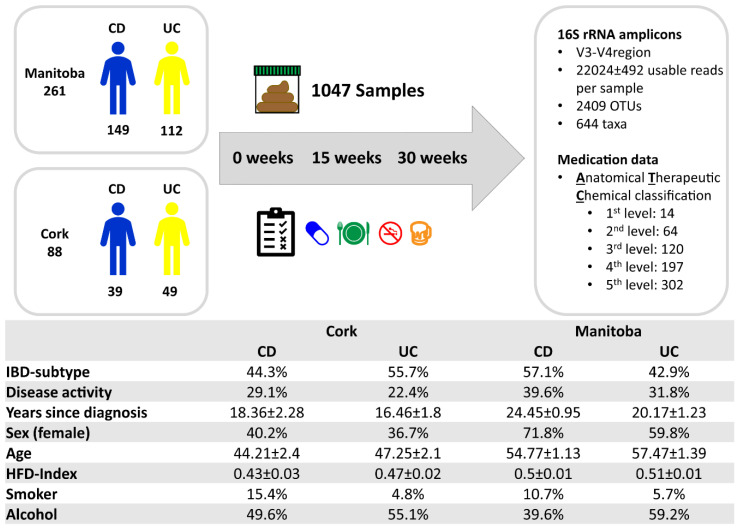
Subject characteristics and sample sizes of the study cohort. The ATC classification system classifies medications hierarchically based on the therapeutic use of the main active ingredient: 1st level: anatomical main group; 2nd level: therapeutic subgroup; 3rd level: pharmacological subgroup; 4th level: chemical subgroup; 5th level: chemical substance.

**Figure 2 microorganisms-10-01963-f002:**
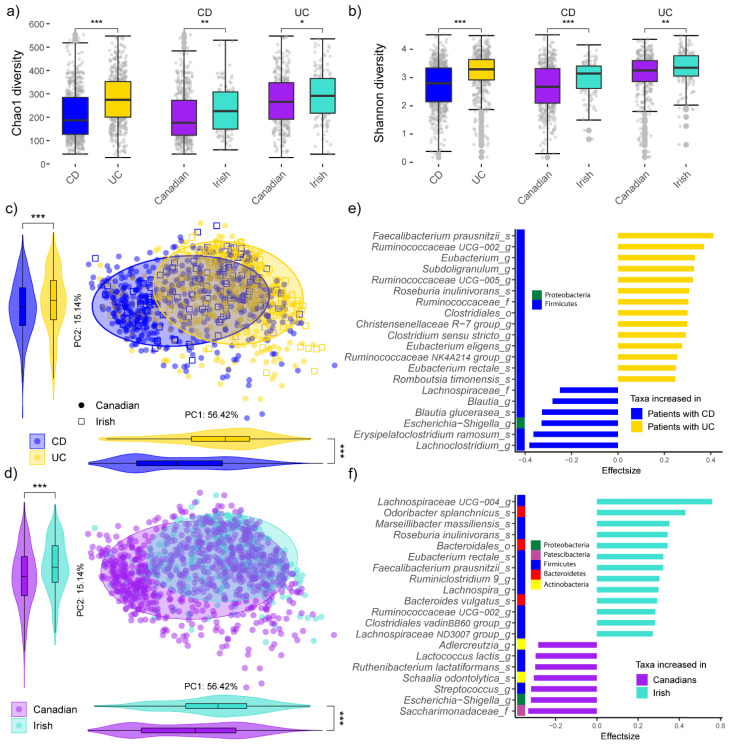
Comparison of (**a**) Chao1 (species richness) and (**b**) Shannon diversity (species richness and evenness) between different IBD-subtypes and geographic location. Principal component analysis (PCA) based on Aitchison distances on all operational taxonomic units (OTUs) present in >10% of samples, with samples grouped by: (**c**) IBD-subtype; and (**d**) geographic location. Violin plots show projections of the PCA points onto PC1 and PC2. Stars show significant differences between the groups as determined by Wilcoxon test. The top 20 most differential OTU abundances between (**e**) IBD-subtypes; and (**f**) geographic location calculated with ALDEx2; * *p* < 0.05; ** *p* < 0.01; *** *p* < 0.001.

**Figure 3 microorganisms-10-01963-f003:**
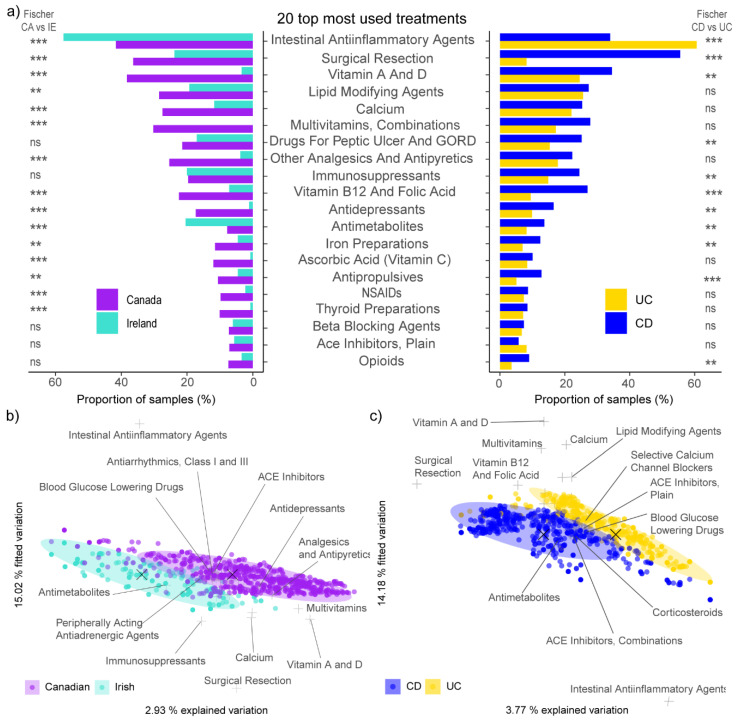
Comparison of the 20 most common medications and supplements taken in the study cohort separated by IBD-subtype and geographic location (**a**). Ordination plot of Jaccard distance-based redundancy analysis (dbRDA) of used medications and supplements constrained by (**b**) geographic location and (**c**) IBD-subtype. The points represent samples, crosses represent medications and X represent the centroids of the depicted groups. The 5% of medications with the best axis fit are labelled; Fishers test: ns *p* > 0.05; ** *p* < 0.01; *** *p* < 0.001.

**Figure 4 microorganisms-10-01963-f004:**
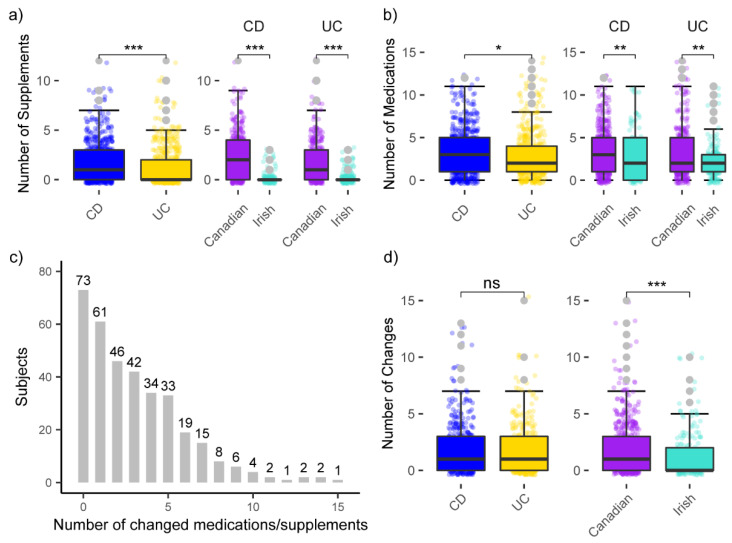
Comparison of the amount of (**a**) supplements and (**b**) medication taken per person and time point; (**c**) histogram of changes in medications and supplements with ATC classification per subject over the course of the study; (**d**) comparison of changes in medications and supplements between persons with differing IBD-subtype and from different geographic locations; Wilcoxon test: ns *p* > 0.05; * *p* < 0.05; ** *p* < 0.01; *** *p* < 0.001.

**Figure 5 microorganisms-10-01963-f005:**
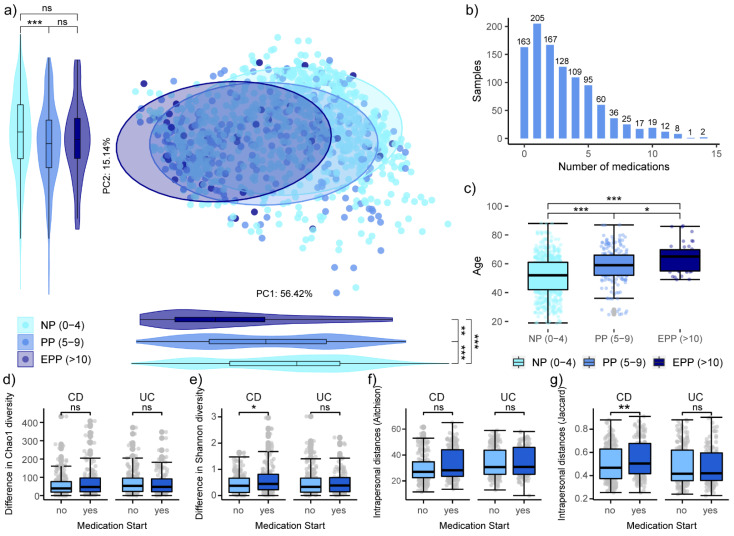
(**a**) PCA based on Aitchison distances with CLR transformed OTUS with a prevalence of at least 10% grouped by polypharmacy. Violin plots show projections of the PCA points onto PC1 and PC2; (**b**) histogram of the number of medications used per sample; (**c**) comparison of the age of participants between the polypharmacy groups. Comparison of intra-personal: (**d**) Chao1 diversity; and (**e**) Shannon diversity differences, as well as (**f**) Aitchison; and (**g**) Jaccard distances. Stars show significant differences between the groups as determined by Wilcoxon test; ns *p* > 0.05; * *p* < 0.05; ** *p* < 0.01; *** *p* < 0.001.

**Figure 6 microorganisms-10-01963-f006:**
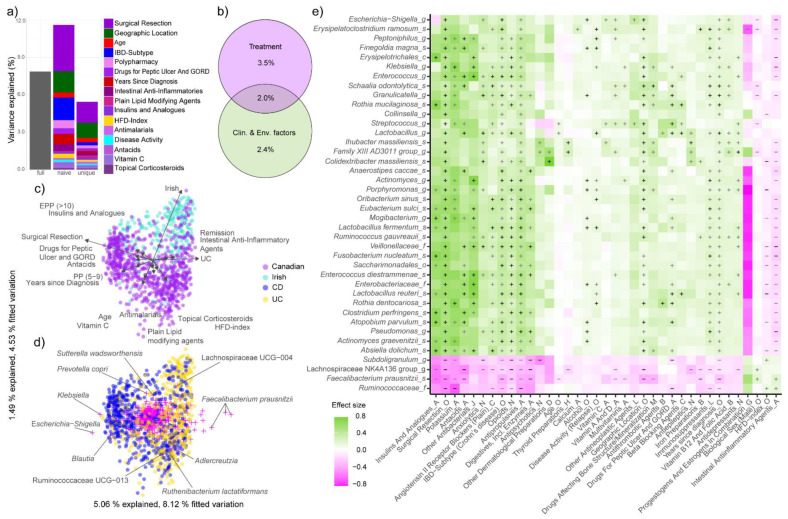
Variance explained (%) by (**a**) individual variables and (**b**) variable groups; ”full” denotes the explained variance for all 16 factors combined, “naïve” the explained variance for each variable separately and “unique” the explained variance for each variable after removing the interaction effects with all other variables; (**c**) biplot of the dbRDA based on Aitchison distances on all operational taxonomic units (OTUs) present in >10% of samples, colored by geographic location. The arrows represent the effects of the constraining variables; (**d**) same dbRDA but colored by IBD-subtype. Crosses denote species sites, and the labels represent the top 10% of OTUs with the best axis fit; (**e**) the top 40 taxa with the highest number of significant associations to metadata. Plus denotes positive associations, minus negative associations. Black signs show strictly deconfounded associations (i.e.; the association can be reduced to this metadata variable/taxonomic feature pair even if another metadata variable is associated to the same feature), grey signs confounded ones (association is distorted by one or more other variables). Taxonomic rank: s: species, g: genus, f: family, c: class, o: order; ATC-levels: A: alimentary tract and metabolism; B: blood and blood forming organs; C: cardiovascular system; D: dermatologicals; G: genito and urinary system and sex hormones; H: systemic hormonal, excl. sex hormones and insulins; L: antineoplastic and immunomodulating agents; M: musculo-skeletal system; N: nervous system; O: other metadata.

**Figure 7 microorganisms-10-01963-f007:**
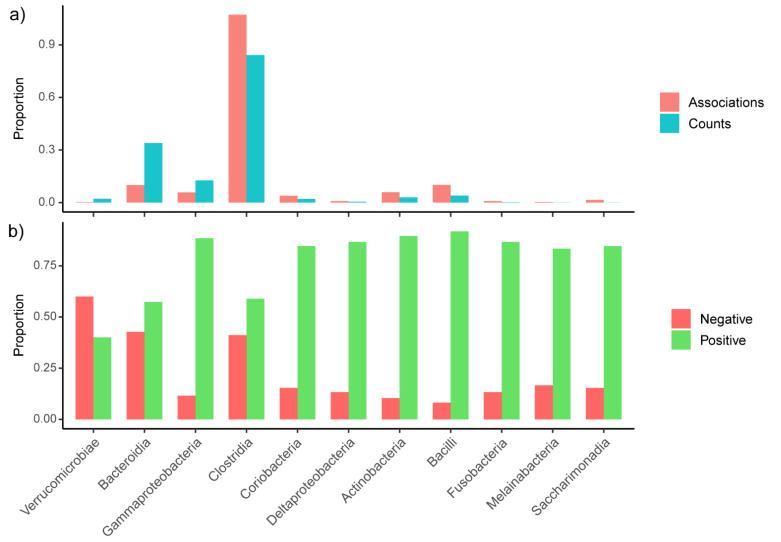
Taxa whose proportions of associations were significantly over- or under-represented compared to the overall distribution of OTU-counts at: (**a**) class; and (**c**) family level as determined by Fisher exact test; proportion of positive and negative associations for: (**b**) class level; and (**d**) family level. Bars are ordered by the ratio of associations to counts and negative to positive associations, respectively.

**Figure 8 microorganisms-10-01963-f008:**
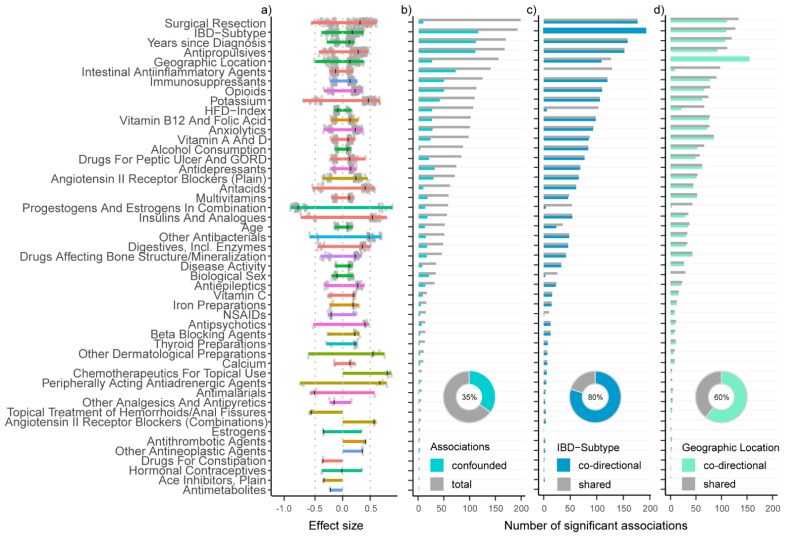
Effect sizes (**a**) and number of significant associations by metadata variable that are (**b**) confounded: (**c**) shared/co-directional with disease; and (**d**) shared/co-directional with geographic location.

## Data Availability

The data presented in this study are openly available in FigShare at doi.org/10.6084/m9.figshare.21088885.v1 and doi.org/10.6084/m9.figshare.20701291.v1. Sequence data are available at NCBI SRA PRJNA414072.

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
