# Peer review of "Interactions between Medications and the Gut Microbiome in Inflammatory Bowel Disease"

_microorganisms, 2022, doi:10.3390/microorganisms10101963_

Round 1
Reviewer 1 Report
“Interactions between medications and the gut microbiome in inflammatory bowel disease” is an original manuscript underlining the differences in the gut microbiota composition of patients affected by CD and UC depending on the drugs administered and on the geographic location. In particular, it is an extension of the study of Clooney et al. and it is successful in describing differences based on medication profiles in more detail compared to the previous one and with a “compositionally-aware analysis” approach.
The study is interesting, but the presentation of data and analysis about treatments should be improved, in order for the manuscript to be more clear and insightful for readers.
Major points
The authors include surgical resection as a treatment. This choice is described in the results section (line 158) and not in the methods. Please describe this approach also in the methods, providing references and rationale for this choice.
Please provide more details on how the raw medication data was processed in the methods (lines 61-62) and include in the results a table with the drugs taken by the patients, the amount and the frequency.
Additional points
Abstract
Results about treatments are presented in a very general and aspecific way in the abstract. Please provide more details on differences in treatments, adding also quantitative details.
Introduction
Lines 45 - 51 of the introduction actually describe results; since these are explained afterwards in the manuscript, please remove this part from the introduction and stress a bit more about the scientific literature regarding changes of microbiota composition depending on administration of drugs, maybe the more commonly used in IBD.
Methods
Please explaine the choice of using 16S rRNA sequencing, instead of whole shotgun metagenomic sequencing, or adding the limit of the methodology chosen (Durazzi et al. 2021) in the discussion part.
Line 66, please provide a reference for the use of fecal calrptectin 250 µg/g as a cut off for active IBD.
Results
Line 110-111, "In order to reduce dimensionality the analysis was conducted at the 3rd ATC level" is unclear, please provide more details also in the methods.
Line 121, please mention in the results the interpretation of alpha and beta diversity, this would be of benefit for readers not expert in the topic
Line 123: "CLR-transformed" is unclear, please provide more details
Line 156, "at level 3 of the ATC classification..." this whole sentence is unclear please revise, adding also in the methods details to better interpret these analysis and results
Discussion
To point out the most interesting results of the manuscript related to the effect of the treatment on the intestinal microflora, it would be nice to deepen and focus more the results and the discussion on the impact of drugs closely related to IBD therapeutic armamentarium (corticosteroids and immunosuppressants for instance). Biologic drugs, such as anti-TNFα, were included in the analysis? Since their role in IBD therapeutic armamentarium is fundamental, it would be correct to spend some words about the contribution of these drugs in the results of the paper and what is available in the scientific literature.
Figure 1: the medication data part, with ATC classification results presented as 1st level, 2nd level etc... is unclear please revise provide more details, with more information on the drug classes involved
Figure 2, please mention in the figure legend interpretation of the Chao1 and Shannon diversity indexes, this would be of benefit for readers not expert in the topic
Author Response
Reviewer 1.1. The authors include surgical resection as a treatment. This choice is described in the results section (line 158) and not in the methods. Please describe this approach also in the methods, providing references and rationale for this choice.
RESPONSE We now describe this in the method section in lines 100-104: “While surgical resection is neither medication or supplement, it is a common treatment for IBD as 80% of patients with CD require surgery during their lifetime [35]. It is also has been shown to significantly alter the microbiome of patients [36] and was therefore included in the analysis.”
Reviewer 1.2. Please provide more details on how the raw medication data was processed in the methods (lines 61-62) and include in the results a table with the drugs taken by the patients, the amount and the frequency.
RESPONSE We now added an explanation of the anatomical therapeutic chemical classification system in the method section in lines 96-100: “The raw medication data was classified based on the anatomical therapeutic chemical (ATC) classification system, which hierarchically classifies medications and dietary supplements based on the therapeutic use of their main active ingredient (1st level: anatomical main group; 2nd level: therapeutic subgroup; 3rd level: pharmacological subgroup; 4th level: chemical subgroup; 5th level: chemical substance) [34], and recorded as a qualitative variable.”
A table of the raw medication data with drugs, dose and frequency (if available) and their conversion into ATC codes is stored on figshare, and will be openly available as soon as the manuscript is published. Supplementary table 2 gives information about which chemical substances were taken by the study cohort and into which pharmacological subgroup they were grouped.
Reviewer 1.3. Results about treatments are presented in a very general and unspecific way in the abstract. Please provide more details on differences in treatments, adding also quantitative details.
RESPONSE: Due to the word limit of 200 words we are not able to present the results in depth, but we now included some more details about the differences in treatment and added some quantitative values to the abstract: “The medication profiles between patients with UC and CD and from different countries varied in number and type of drugs taken. Among Canadian patients, surgical resection, and overall medication and supplement usage is significantly more common than for their Irish counterparts. Treatments explained more microbiota variance (3.5%) than all other factors combined (2.4%) and 40 of the 78 tested medications and supplements showed significant associations with at least one taxon in the gut microbiota.”
Reviewer 1.4. Lines 45 - 51 of the introduction actually describe results; since these are explained afterwards in the manuscript, please remove this part from the introduction and stress a bit more about the scientific literature regarding changes of microbiota composition depending on administration of drugs, maybe the more commonly used in IBD.
RESPONSE: We expanded the introduction to include known effects of IBD and non-IBD drugs on the microbial composition and vice versa, please see lines 44-71 in the manuscript. Respectfully, we have decided to keep the final paragraph as an appropriate prelude and brief summary of the major findings
Reviewer 1.5. Please explain the choice of using 16S rRNA sequencing, instead of whole shotgun metagenomic sequencing, or adding the limit of the methodology chosen (Durazzi et al. 2021) in the discussion part.
RESPONSE We revisited a previous study based on the widely utilized and well-established method of 16S rRNA amplicon sequencing. While this method offers a lower taxonomic resolution compared to shotgun metagenomics and no functional information, its cost effectiveness allows meaningful interrogation of these large sample sizes.
Reviewer 1.6. Line 66, please provide a reference for the use of fecal calprotectin 250 µg/g as a cut off for active IBD.
RESPONSE The reference is now added “Lin, J.F., et al., Meta-analysis: fecal calprotectin for assessment of inflammatory bowel disease activity. Inflamm Bowel Dis, 2014. 20(8): p. 1407-15.”
Reviewer 1.7. Line 110-111, "In order to reduce dimensionality the analysis was conducted at the 3rd ATC level" is unclear, please provide more details also in the methods.
RESPONSE We have now clarified this in lines 153-158: “Applying the anatomical therapeutic chemical (ATC) classification system [34] to the raw medication data yielded 302 different chemical substances (5th ATC level). [..] For the analysis the medications and supplements were then further combined into 120 pharmacological subgroups (3rd ATC level) and drug usage was recorded as a qualitative yes-no variable.”
Reviewer 1.8. Line 121, please mention in the results the interpretation of alpha and beta diversity, this would be of benefit for readers not expert in the topic
RESPONSE We added a short explanation of alpha and beta diversity in the text in line 171 and 174: “As previously reported [29], alpha (i.e. within sample ) diversities were lower for all patients with CD compared to UC, and also for Canadian participants in general (Wilcox p < 0.05; Figure 2a and b, Suppl. Table 3). Abundances of OTUs were CLR-transformed and beta (i.e. between sample) diversity analysis was calculated from Aitchison distances of all OTUs present in at least 10% of the samples.”
Reviewer 1.9. Line 123: "CLR-transformed" is unclear, please provide more details
We now explain the centred log ratio (CLR) transformation and its use in the method section in lines 110-115: ”In order to alleviate the constant sum constraint of compositional data [39], zeros were removed from the raw counts via the count zero multiplicative method within the zCompositions_1.3.4 package [40] and subsequently subjected to a centred log ratio (CLR) transform, i.e. the data were expressed as logarithms of ratios with the geometric mean as denominator using the propr_4.2.6 package [41].”
Reviewer 1.10. Line 156, "at level 3 of the ATC classification..." this whole sentence is unclear please revise, adding also in the methods details to better interpret these analysis and results.
RESPONSE: We now clarified this sentence in lines 207-208: “Of the 120 different pharmacological subgroups found in this study, 78 were recorded at least 5 times as “yes” in total and were included in the downstream analysis (default MetadeconfoundR requirement).”
Reviewer 1.11. To point out the most interesting results of the manuscript related to the effect of the treatment on the intestinal microflora, it would be nice to deepen and focus more the results and the discussion on the impact of drugs closely related to IBD therapeutic armamentarium (corticosteroids and immunosuppressants for instance). Biologic drugs, such as anti-TNFα, were included in the analysis? Since their role in IBD therapeutic armamentarium is fundamental, it would be correct to spend some words about the contribution of these drugs in the results of the paper and what is available in the scientific literature.
RESPONSE:
We agree that medications that are used to treat IBD, which are in this study mainly included in the subgroups “intestinal anti-inflammatory agents” and “immunosuppressants”, and their important effect on the intestinal flora are now specifically mentioned both in the results in lines 391-398: “For example, the immunosuppressant subgroup (L04A), which contains selective immunosuppressants such as mycophenolic acid but also tumor necrosis factor alpha (TNFα) inhibitors and other immunosuppressants like methotrexate and azathioprine, showed significant associations with 124 taxonomic features. Of these, 120 were not only also significant for IBD-subtype but co-directional as well. A notable exemption to this exacerbating effect were intestinal anti-inflammatories, a pharmacological subgroup that contains locally acting corticosteroids, and aminosalicylic acid and similar agents and shared 129 of its 140 significant associations with IBD-subtype, but none of those were co-directional, suggesting that they counteracted the effect of IBD-subtype towards a healthier microbiome.” and in the discussion in lines 447-453 “The lack of diminution of this taxon by immunosuppressants here, might be explainable due to the fact that the ATC subgroup L04A not only includes thiopurines but also TNFα inhibitors, latter of which have been shown to increase SCFA producing bacteria [58]. It is noticeable though that immunosuppressants share nearly all their significant associations to taxonomic features with IBD subtype and all of those shared associations showed the same directionality and thus increases the disparity between the gut microbiota of persons with UC and CD. In contrast to that, intestinal anti-inflammatory agents are among the few drugs that did not follow this exacerbating trend. While this medication subgroup also shared most of its significant associations to taxa with IBD-subtype, none of them were co-directional. This observation is in agreement with earlier reports that 5-ASA drugs can partially recover the gut microbiome to a healthy status [29, 59].”
On the other hand, as IBD medications were already discussed in our previous study and due to the multitude of other non-IBD medications and supplements that are used by the study participants, we think that our focus shouldn’t be constrained on the impact of drugs that are closely related to IBD but also on the confounding effects of the additionally taken other medications that have been shown to impact the gut microbial composition and probably contribute to the conflicting results across different IBD studies. To bring this point better across, we added in line 472: “Furthermore, it highlights the need to include an exhaustive list of medication intake (and ideally dosages) of study participants in the analysis that go beyond the common IBD therapeutics to improve reproducibility between IBD-studies.”
Reviewer 1.12. Figure 1: the medication data part, with ATC classification results presented as 1st level, 2nd level etc... is unclear please revise provide more details, with more information on the drug classes involved
RESPONSE: We added an explanation of the meaning of the different ATC levels to the figure caption: “The ATC classification system classifies medications hierarchically based on the therapeutic use of the main active ingredient: 1st level: anatomical main group; 2nd level: therapeutic subgroup; 3rd level: pharmacological subgroup; 4th level: chemical subgroup; 5th level: chemical substance.” As there are 302 chemical substances at the 5th ATC level and 120 different pharmacological subgroups at the 3rd ATC level, giving more information on the involved drug classes is not feasible in the figure caption. The information is available in Supplementary tables 1 and 2 and, in part, also in Figure 3.
Reviewer 1.13. Figure 2, please mention in the figure legend interpretation of the Chao1 and Shannon diversity indexes, this would be of benefit for readers not expert in the topic
RESPONSE: We added an interpretation for Chao1 and Shannon diversity and the figure caption now reads: “Comparison of (a) Chao1 (species richness) and (b) Shannon diversity (species richness and evenness) between different IBD-subtypes and geographic location.”
Reviewer 2 Report
The submitted manuscript presents data on interactions between medications and the gut microbiome in IBD cohort. Within this study additional analyses of already published data is presented with subset of data used and further/different analyses performed with the aim to thoroughly study the effects of different treatments (surgical, pharmacological, dietary supplements) on gut microbiota in IBD subsets. The only substantial complaint is that a lot of data on medications has been published in previous manuscript and the novelty of the work is not high. The manuscript is well structured and written. The authors provide detailed methodology descriptions and present the data, results and discussion thoroughly and clearly. The results of this work are of high interest to the scientists from the field.
Before publishing only some minor revisions are advised.
1. The data on sampling time points reported here differs from the data reported in reference 6 (Clooney AG, et al. Ranking microbiome variance in inflammatory bowel disease: a large longitudinal intercontinental study. Gut. 2021 Mar;70(3):499-510) i.e. 16 weeks versus 3 months. The error or discrepancy needs to be corrected or explained.
2. The information in lines 156-157 that “78 medications and supplements were taken at 5 more sampling time points” is unclear or incorrect. It is understood that there were 3 sampling time points so this needs to be corrected or clarified. Overall, the use of term “time point” is somewhat confusing so there should be some clarification in that respect throughout the manuscript.
3. It seems that at this point of submission the Figures are of very low quality and this needs to be improved upon re-submission.
4. There are many references in the manuscript to supplementary data but this data could not be accessed or reviewed as the link in the manuscript reports error – file not found.
Author Response
Reviewer 2.1: The data on sampling time points reported here differs from the data reported in reference 6 (Clooney AG, et al. Ranking microbiome variance in inflammatory bowel disease: a large longitudinal intercontinental study. Gut. 2021 Mar;70(3):499-510) i.e. 16 weeks versus 3 months. The error or discrepancy needs to be corrected or explained.
RESPONSE: We apologize for this error, the actual sampling time points are on average 15.18±0.42 weeks apart. The difference to the previous study can be explained by the different subset of samples. The times were now corrected in figure 1 as well as in the manuscript in line 149-150: “The sampling time points were on average 15.18±0.42 weeks apart.”
Reviewer 2.2: The information in lines 156-157 that “78 medications and supplements were taken at 5 more sampling time points” is unclear or incorrect. It is understood that there were 3 sampling time points so this needs to be corrected or clarified. Overall, the use of term “time point” is somewhat confusing so there should be some clarification in that respect throughout the manuscript.
RESPONSE: As each patients gave 3 samples, a medication or supplement that is recorded as “yes” 5 times throughout the study doesn’t necessarily mean that 5 different patients took the drug at one time during the study but could also mean that 2-4 patients took the drug at multiple time points which was to be expressed by “5 more sampling time points” but as this is indeed a vague/confusing expression we changed it in line 207 to: “Of the 120 different pharmacological subgroups found in this study, 78 were recorded at least 5 times as “yes” in total and were included in the downstream analysis (default MetadeconfoundR requirement).” And in line 343: “We included 79 treatments and supplements, which were recorded at least 5 times as “yes” within the study, [..]”
Reviewer 2.3: It seems that at this point of submission the Figures are of very low quality and this needs to be improved upon re-submission.
RESPONSE: At the point of submission PNGs of the figures were included in the word document, while high quality TIFFs were submitted separately. We now exchanged the figures in the word document to TIFFs.
Reviewer 2.3: There are many references in the manuscript to supplementary data but this data could not be accessed or reviewed as the link in the manuscript reports error – file not found
RESPONSE: The raw medication data and meta data for this study are stored on figshare will be made available as soon as the paper is published. A separate file with supplementary data was submitted.